# Supergene regulation of ant social organization: a *P* haplotype in workers shifts colony ontogeny towards multiple queens

Ornela De Gasperin [1,2] ✉, Pierre Blacher[2], Marina Choppin[2] & Michel Chapuisat [2] ✉

Supergenes are large clusters of linked genes that control complex phenotypes. In several ant clades, supergenes determine whether one or multiple queens reproduce in mature colonies, but how supergenes affect colony social structure has been inferred indirectly. We show experimentally that a supergene in *Formica* ant workers alters the social structure (single- or multi-queen) of developing colonies. We crossed queens and males carrying alternative social supergene genotypes, and let the queens establish single-queen colonies, which we paired in the laboratory. The presence of a paternally-inherited *P* haplotype in workers was sufficient to make the colonies more likely to fuse and become multi-queened, regardless of the genotype of their mother. The dominant effect of the *P* haplotype on colony social structure likely contributes to the spread of multi-queen colonies. This controlled experiment provides direct evidence that the *P* haplotype in workers steers social organization towards multiple queens during early colony ontogeny.

Ant societies vary widely in size and organization. They can host up to several million individuals, with many non-breeding workers helping one to many breeding queens[1–3]. The ontogenetic specialization of castes in ants resembles the ontogenetic differentiation of cell lineages in metazoans, with the term 'superorganism' used to highlight the fundamental differences between most ant species and other animal societies (where individuals remain reproductively and behaviorally totipotent)[4,5]. The ancestral life cycle of ants, present in species that have a single queen per colony (monogyne), starts with a young queen that has flown away from her maternal nest, mated in a swarm, and founded her new colony alone[1,6]. The queen raises a first cohort of daughter-workers that forage and help raise sister-workers. As the incipient colony grows, the new workers produced are physically, behaviorally, and physiologically different from the first cohort of workers[1,7]. These individual-level phenotypic differences lead to colony-level phenotypic changes, such as colonies switching from producing workers to raising young queens and males.

Some ant species harbor two types of colonies, with either one or multiple reproducing queens per colony[1,8,9]. In at least five independent ant lineages, this intraspecific variation in colony social structure (one or multiple queens per colony) is controlled by large genomic regions of low recombination, called supergenes[10,11]. The best-studied cases are the *Solenopsis* and *Formica* genera, which harbor independently evolved 'social supergenes' (supergenes that underlie queen number and other traits associated with ant social organization)[12–14]. In both systems, all individuals living in mature single-queen colonies exclusively carry the ancestral, non-inverted supergene haplotype (*B* and *M*, respectively). In contrast, some or all of the ants living in mature multi-queen colonies carry one or two copies of the derived, inverted haplotype (*b* and *P*, respectively)[10–12,15]. In both clades, mature monogyne and polygyne colonies differ in many aspects, including colony size, lifespan, and sex allocation[8,9,16], as well as in individual body size and behavior[17–21].

The way by which the derived haplotype of the supergene affects queen and worker behavior to promote polygyny remains little understood. In fire ants, *S. invicta*, workers that lack the haplotype associated with polygyny kill supernumerary queens, while workers carrying this haplotype accept extra queens if they also carry this haplotype[22,23]. Moreover, it has been suggested that monogyne queens mating with polygyne males can spread the polygyne social organization[24]. In Alpine silver ants, *F. selysi*, most young queens disperse on the wing, join mating swarms[25], and are rejected by unrelated colonies[26,27]. Queens originating from polygyne colonies, which carry the derived supergene haplotype, occasionally mate with males of monogyne origin, but the majority mate assortatively, with males of polygyne origin[25,28].

[1]Red de Ecoetología, Instituto de Ecología A.C., Instituto de Ecología A.C., Xalapa, Veracruz, México. [2]Department of Ecology and Evolution, University of Lausanne, Lausanne, Switzerland. ✉e-mail: ornela.degasperin@inecol.mx; michel.chapuisat@unil.ch

Queens of polygyne origin are prone to starting a nest cooperatively, and these associations persist after worker emergence[29]. Furthermore, polygyne colonies tend to accept additional daughter queens as colonies grow[27,28]. In contrast, queens originating from monogyne colonies, which lack the derived haplotype, start their nest alone[29] and typically establish colonies that remain single-queened throughout their estimated ten-year lifespan[9]. Queens of monogyne origin occasionally mate with males of polygyne origin, resulting in incipient colonies in which the founding queen and her daughter carry supergene genotypes inducing alternative forms of colony social organization[25,29].

Overall, the social supergene of *F. selysi* affects the developmental trajectory of the colony, with the derived haplotype steering colony organization towards multiple queening. However, because queens and workers with each of the alternative supergene genotypes typically develop in separate colonies that differ in social organization and many associated phenotypic traits[9,16–19], experiments so far failed to distinguish direct genetic effects of the supergene from indirect effects mediated by the social environment.

Here, we investigated the direct genetic effect of the supergene on the ontogeny of incipient colonies of Alpine silver ants, *F. selysi*. This species, native to south-west Europe and non-invasive, has monogyne and polygyne colonies throughout the Alpine range[30]. Colony social structure (one or multiple queens) is associated with two haplotypes of a large social supergene, *M* and *P*, *M* being the ancestral haplotype and *P* the derived, inverted haplotype associated with polygyny[12,13]. *F. selysi* queens usually mate once in the field, especially in monogyne colonies[28]. All mature colonies typically contain either a single *MM* queen mated to an *M* male (=queen having the *MM* supergene genotype, mated to a male carrying the *M* haplotype; ants have diploid females and haploid males) or multiple queens carrying at least one copy of the *P* haplotype (=queens having the *MP* or *PP* supergene genotypes)[12,28,31]. Field data and laboratory experiments revealed rare events of joint colony founding by multiple queens and fusion of small incipient colonies[26,29]. Genetic data showed that nestmate queens are as related as workers, indicating that acceptance of nestmate queens is common in polygyne colonies[28].

Queens in polygyne colonies can be mated to *M* or *P* males[28,32], but they produce workers, young queens, and males with at least one copy of the *P* haplotype, because brood from *MP* queens that lacks the *P* haplotype fails to develop[31]. Around 20% of *MM* queens in swarms mate with *P* males[25], a cross that yields viable colonies in the laboratory[32–34] and that establishes incipient colonies in the wild[29], but that has never been observed in mature colonies, despite extensive monitoring[12,25,28]. Specifically, we have never detected an *MM* queen mated to a *P* male in mature colonies, neither in monogyne nor in polygyne colonies, although for the latter our sample size was limited[12,25,28]. As all workers from this cross inherit the *P* haplotype, the growing colony may tolerate extra queens and become multi-queened, which is what *P*-carrying workers do in polygyne colonies. This contrasts with the single-queen structure expected based on the supergene genotype

of the founding queen. The single founding *MM* queen mated to a *P* male may then die and be replaced by her *MP* daughter-queens.

To test if the supergene alters colony social structure by directly affecting either workers or queens, we experimentally controlled the supergene genotype of each caste in incipient single-queen colonies. We crossed queens and males of known social origin and allowed each queen to establish an incipient colony in the laboratory. We then paired single-queen incipient colonies of known supergene composition and assessed their tendency to fuse. Unlike field experiments or behavior assessments of field-caught females, where supergenes may have indirect genetic effects on other group members (for instance, if the supergene of queens indirectly affects the behavior of their offspring), this controlled approach with incipient colonies established by a single *MM* queen crossed with either an *M* or a *P* male allowed us to assess the direct genetic effect of a paternally-inherited supergene in workers. We found that colony fusion indeed varied according to the social supergenes of colony members, and that a paternally-inherited *P* haplotype in workers was sufficient to shift colony development towards multiple queening. These results provide direct experimental evidence that the *P* haplotype present in *Formica* ant workers influences the social structure of their colony during its ontogeny.

## Methods
### Experimental approach
We crossed *F. selysi* queens and males originating from each social form, resulting in three types of incipient single-queen colonies: queens with monogyne supergene genotypes mated to males with monogyne supergene haplotypes (*MM* queens with *MM* workers, expected to develop into monogyne colonies), queens with polygyne supergene genotypes mated to either *M* or *P* males (*MP* or *PP* queens and *MP* or *PP* workers expected to develop into polygyne colonies), and queens with monogyne supergene genotypes mated to males with polygyne supergene haplotypes (producing a colony with 'mixed' supergene genotypes, i.e. *MM* queens with *MP* workers; hence, colony ontogeny will depend on the respective effects of the supergene on workers and queens). After one year, we measured colony size and paired colonies of all types (monogyne, polygyne or mixed supergene genotypes). We let paired colonies interact and recorded whether the two queens survived or not, and whether the colonies merged into a multi-queen colony or not. We predicted that colonies with monogyne supergene genotypes would remain single-queened and would frequently kill other queens, while colonies with polygyne supergene genotypes and colonies with mixed genotypes would be less likely to kill other queens and more likely to form multi-queen colonies.

### Field collection, artificial mating and independent colony founding
Details of the field collection and artificial mating procedures can be found in refs. 33,35. Briefly, we collected sexual pupae and workers in Valais, Switzerland, in summer 2017 and 2019. We determined the social

**Table 1 | Total number of replicates per treatment (pairs of colonies with mixed, monogyne, or polygyne supergene genotypes, respectively), separated by experimental year and colony age**

| Treatment | | Total | Year | | | Age of colonies | | |
|---|---|---|---|---|---|---|---|---|
| Supergene genotypes in colony 1 | Supergene genotypes in colony 2 | | 2018 | 2019 | 2020 | 1-year-old | 2-years-old | 3-years-old- |
| Mixed (*MM* queen, *MP* workers) | Mixed (*MM* queen, *MP* workers) | 53 | 20 | 17 | 16 | 30 | 17 | 6 |
| Mixed (*MM* queen, *MP* workers) | Monogyne (*MM* queen, *MM* workers) | 31 | 15 | 16 | 0 | 15 | 16 | 0 |
| Mixed (*MM* queen, *MP* workers) | Polygyne (*MP* or *PP* queen, *MP* or *PP* workers) | 32 | 17 | 14 | 1 | 17 | 14 | 1 |
| Monogyne (*MM* queen, *MM* workers) | Monogyne (*MM* queen, *MM* workers) | 9 | 2 | 0 | 7 | 3 | 0 | 6 |
| Monogyne (*MM* queen, *MM* workers) | Polygyne (*MP* or *PP* queen, *MP* or *PP* workers) | 7 | 7 | 0 | 0 | 7 | 0 | 0 |
| Polygyne (*MP* or *PP* queen, *MP* or *PP* workers) | Polygyne (*MP* or *PP* queen, *MP* or *PP* workers) | 14 | 11 | 0 | 3 | 12 | 0 | 2 |

organization and supergene genotypes of each field colony by SNP geno-typing three individuals per colony[12,25]. We used young queens originating from 43 and 23 monogyne and polygyne field colonies, respectively. We let young queens mate with non-nestmate virgin males of known social origin (and thus, supergene haplotype). We then allowed each queen to start her own colony alone. These colonies went through an experiment that mimicked mild or harsh winters, published in ref. 33. After that experiment ended (at least one year after mating) we placed these colonies in the experiment described below (Table 1).

### Experimental design

We performed this experiment throughout three consecutive summers. Each replicate consisted of a pair of colonies, each with a single queen, each within its own box (15.5 ×13.5 ×5.5 cm), united through a smaller plastic box (10.5 ×13.5 ×5.5 cm) placed between them and connected through rubber tubes (ø = 5 mm; length 4 cm), as described in ref. 26. Inside each colony box we provided water *ad libitum* in a glass test tube (length = 16 cm; ø = 5 mm) one-third filled with water. We kept these boxes in standard laboratory conditions, at 25 °C, 70% humidity, and in a light:dark 12:12 h cycle. We fed colonies twice a week with egg and apple jelly, provided *ad libitum*, placed in the smaller plastic box between both colonies (so workers and/or queens from both colonies had to forage for food in the central common area). All boxes were side-lined with fluon to prevent ants from escaping. We tested all colony combinations in a total of 146 pairs of colonies, maximizing replicates containing colonies with "mixed" supergene genotypes (Table 1). Out of 227 single-queen laboratory colonies, 167 were used only once, and 60 were used more than once. In the statistical analyses, we controlled for having used some laboratory colonies more than once in the behavioral experiments, and for having queens emerging from the same field colonies. We paired same-age colonies (except for one pair), to minimize differences in brood size between pairs of colonies, but matching was done randomly with respect to worker number. Note that the median size of one-year-old colonies was six workers, and that of two-year-old colonies was 199.

Two weeks before starting the experiment, we counted the total number of workers and pupae in each colony and paint-marked all queens for identification (workers were not individually marked). Colonies were not producing sexual brood, and eggs and larvae were not counted. Once we allowed colonies to interact, we performed observations twice a week during the first month, and then once a week for at least another month, as described in ref. 26. We recorded whether the two queens were alive, and their spatial location (if they were in their own nest, in the other colony's nest, or in the central common area). As the queens were painted either green or white (randomly), an observer blind to the color code recorded which of the two queens died. We categorized the outcome as a 'multi-queen colony' when both queens cohabited in the same nest most times, following[26]. The experiment was conducted blindly to the supergene genotypes of the colonies.

### Statistical analyses

All analyses were carried out in R v.3.5.1[36], using generalized linear mixed effects models (GLMM) with the lme4[37] or nlme package[38], or with Firth's bias-reduced logistic regression (logistf package[39,40]). We obtained model estimates with type II SS ('Anova' function[41]), and estimates, SE and p values with the 'summary' function[36]. We did not include interactions due to small sample sizes of some pairs of colony types (see Table 1). We did *post hoc* comparisons with FDR p-value adjustment ('lsmeans' function[42]). We evaluated model assumptions following[43].

### Colony size

We compared the size of colonies (workers plus pupae) with monogyne, polygyne, and "mixed" supergene genotypes when first used in the experiment (when colonies were one year old), that is before they could gain or lose workers while being paired with another colony. We compared the size of 168 one-year-old colonies (28, 42 and 98 with monogyne, polygyne

and "mixed" supergene genotypes, respectively), using a GLMM with Poisson error distribution. We included an observation level as a random effect, where each data point receives a unique level of a random effect which models the extra-Poisson variation to control for overdispersion[44]. We included the supergene genotypes of the colonies as an explanatory variable (monogyne, mixed or polygyne), and the field colony of origin of the queens as a random effect, nested within the experimental year (because queens collected from the same field colony within the same year are likely to be more similar to one another). Because we found heteroscedasticity in the residuals, we used the 'lme' function to control for differences in the variance of the residuals between colony genotypes using the *varIdent* variance structure and verified the normalized residuals, following[43].

### Multi-queen colony formation

We predicted that colonies with workers carrying polygyne supergene genotypes would be more likely to fuse with other colonies than colonies with workers carrying monogyne supergene genotypes. We compared the probability that pairs of colonies would become a multi-queen colony with a GLM with binomial error distribution. We included one observation per replicate, with the outcome being 'multi-queen colony' (1), or 'not multi-queen colony' (0). We included the treatment with six levels (monogyne-monogyne/monogyne-mixed/monogyne-polygyne/mixed-mixed/mixed-polygyne/polygyne-polygyne) as an explanatory variable. We did not include the size of each colony as an independent factor because the two values were highly correlated with one another (Spearman's rho = 0.75), as we had paired same-age colonies. Instead, we included as a continuous covariable the sum of the number of workers of the two colonies, to control for the overall number of workers in the replicate. We did not include the experimental year as a random effect, as this variable was highly correlated with the overall size of the two colonies (p < 0.000001). We included as random effects the identity of the two laboratory colonies, to control for having used colonies twice, each nested within the field colony of origin of the queens. However, because most colonies were used only once, the identity of the colonies explained almost all the variance in the response variable, leading to model convergence issues. Therefore, we included an extra variable as a fixed factor, in which we coded if a colony within a replicate had been used before or not (yes/no). We found that the treatment significantly predicted the proportion of multi-queen colonies, but we encountered complete separation between the response and some treatments, which led to very high standard errors, precluding us from carrying out post hoc tests (no replicate in the monogyne-monogyne nor in the monogyne-polygyne treatments ended as a 'multi-queen colony'). To overcome this problem, we ran a Firth's bias-reduced logistic regression (package 'logistf'[39,40]), which adds a Jefrey's prior to allow for analyses when there is complete separation between the response and explanatory variable, but which does not allow to include random effects. Each replicate provided one outcome, multi-queen colony, or not multi-queen colony. We included as explanatory variables the supergene genotype of colony one, and the supergene genotype of colony two within a replicate (to avoid carrying 15 post-hoc comparisons among all treatments), if a laboratory colony within a replicate had been used before or not (yes/no), and the total number of workers in the replicate. We obtained estimates and p values using the summary function. We first ran the model using the monogyne genotype as reference, and then re-ran the model to compare mixed and polygyne genotypes. For pairs of colonies with the same supergene genotypes, it did not matter which colony was 'colony one' and which was 'colony two'. For pairs of colonies with different supergene genotypes (monogyne-mixed, mixed-polygyne, monogyne-polygyne), we randomly allocated half of the colonies of one genotype to 'colony one', and the other half to 'colony two'. To account for the effect of the variation in colony sizes, we re-ran the model with z-scores for the total size of the two colonies. To account for differences in the number of replicates, we re-ran the analysis using only large sample size treatments ('mixed-monogyne', 'mixed-polygyne', and 'mixed-mixed'). We also bootstrapped 1,000,000 times 12 replicates and plotted the distribution of the estimates of the likelihood of multi-queen colony formation

for these three treatments (see Supporting Information). In this and in the following analysis (queen mortality), we excluded one replicate which was lost (the queens escaped through a hole below the rubber tube that united the boxes).

## Queen mortality

We predicted that colonies with workers carrying polygyne supergene genotypes would be less likely to reject the other queen, and less able to defend their own queen, than colonies with workers carrying monogyne supergene genotypes. We compared the probability of death of the queen in a GLMM with binomial error distribution (1 = the queen died). We included the supergene genotypes of her own colony and the supergene genotypes of the second colony as explanatory variables, each with three levels (monogyne/mixed/polygyne). We included the difference in the size (number of workers) of the two colonies as a continuous covariable, as we expected smaller colonies to have more chances of losing their queen. We included the replicate as a random effect, to control for the non-independence of the two queens' mortality within a replicate, nested within the year of the experiment. We also included the identity of the laboratory colony, nested within the field colony of origin of the queens as a second random effect, to control for having used some laboratory colonies more than once, and for having collected some queens from the same field colonies, nested within the year when queens were collected, because queens collected from the same field colony within the same year are likely to be more similar to one another. We re-ran the model with z-scores for the size difference between colonies.

## Reporting summary

Further information on research design is available in the Nature Portfolio Reporting Summary linked to this article.

## Results

### Colony size

Colonies with mixed supergene genotypes, i.e. *MM* queen and *MP* workers, were larger than colonies with only polygyne supergene genotypes, and as large as those with monogyne supergene genotypes ($\chi^2 = 8.62$; $p = 0.01$; $n = 28$, 98, and 42 colonies with monogyne, mixed and polygyne supergene genotypes, respectively; Fig. 1; *post hoc* comparisons: Estimate (monogyne vs mixed) = $-0.36$; SE = 1.12; $t = -0.32$; $p = 0.74$; Estimate (mixed vs polygyne) = 2.82; SE = 0.97; $t = 2.89$; $p = 0.01$; Estimate (monogyne vs polygyne) = 2.45; SE = 1.26; $t = 1.93$; $p = 0.08$).

### Multi-queen colony formation

When workers carried the *P* haplotype, a replicate was significantly more likely to become a multi-queen colony, independently of whether that haplotype was inherited from the father or from the mother. The experimental treatment – the combination of queen and worker supergene genotypes in pairs of single-queen colonies (with six levels) – significantly predicted whether the two colonies fused into a multi-queen colony ($\chi^2 = 19.68$; $p = 0.001$; $n = 9$ monogyne-monogyne, 31 monogyne-mixed, 7 monogyne-polygyne, 53 mixed-mixed, 32 mixed-polygyne, 14 polygyne-polygyne; Fig. 2). When we ran Firth's bias-reduced logistic regression, we found that a replicate was significantly more likely to become a multi-queen colony when workers of both colonies carried the *P* haplotype, as compared to replicates involving one or two colonies with workers carrying only monogyne supergene genotypes (effect of supergene genotypes in colony 1: monogyne vs. mixed: Coef (mixed) = 1.91 with 95% CI [0.37, 4.20]; $\chi^2 = 6.39$, $p = 0.01$ (with z-scored colony sizes: $p = 0.01$); monogyne vs. polygyne: Coef (polygyne) = 1.85 with 95% CI [0.10, 4.24]; $\chi^2 = 4.37$, $p = 0.03$ (with z-scored colony sizes: $p = 0.03$); effect of supergene genotypes in colony 2: monogyne vs. mixed: Coef (mixed) 1.58 with 95% CI [0.11, 3.82]; $\chi^2 = 4.55$, $p = 0.03$ (with z-scored colony sizes: $p = 0.03$); monogyne vs. polygyne: Coef (polygyne) = 2.02 with 95% CI [0.39, 4.34]; $\chi^2 = 6.25$, $p = 0.01$ (with z-scored colony sizes: $p = 0.01$)).

When workers carried a *P* haplotype, the probability that a replicate would become a multi-queen colony was similar regardless of whether the queen had a monogyne or polygyne supergene genotype (effect of supergene genotype of colony 1: mixed vs. polygyne: Coef (polygyne) = 0.05 with 95% CI [$-1.11$, 1.16]; $\chi^2 = 0.0009$, $p = 0.92$ (with z-scored colony sizes: $p = 0.92$); effect of the genotype of colony 2: mixed vs. polygyne: Coef (polygyne) = 0.44 with 95% CI [$-0.56$, 1.43]; $\chi^2 = 0.77$, $p = 0.38$ (with z-scored colony sizes: $p = 0.38$)). Therefore, a single *P* haplotype in workers, inherited through the father or the mother, was sufficient to promote multiple queening. Whether the replicate contained a colony that had been used before or not did not predict whether the replicates would end up as a multi-queen colony (Coef = 0.41 with 95% CI [-0.80, 1.70]; $\chi^2 = 0.45$, $p = 0.50$). The proportion of multi-queen colonies did not depend on the overall number of workers in the replicate ($\chi^2 = 2.65$, $p = 0.10$).

When we re-ran the model using only treatments with large sample sizes, we found similar results. The proportion of multi-queen colonies was smaller in the monogyne-mixed treatment than in both the mixed-mixed treatment (Coef (mixed-mixed) = 1.78 with 95% CI [0.45, 3.50]; $\chi^2 = 7.37$, $p = 0.006$) and the mixed-polygyne treatment (Coef (mixed-polygyne) = 1.47 with 95% CI [0.01, 3.25]; $\chi^2 = 3.90$, $p = 0.04$). The proportion of multi-queen colonies did not differ between the mixed-mixed and the mixed-polygyne treatments (Coef (mixed-polygyne) = -0.30 with 95% CI [$-1.36$, 0.69]; $\chi^2 = 0.34$, $p = 0.55$). The bootstrapped estimates with 12 resampled replicates showed proportions of multi-queen colonies similar to the ones in the full data set (Fig. 2, Supplementary Fig. 1 and Supplementary Fig. 2 in Supporting Information).

### Queen mortality

Overall, queens in colonies containing workers with the paternally-inherited *P* haplotype were more likely to die. Specifically, queens with monogyne supergene genotypes were more likely to die when their own workers had polygyne genotypes, rather than monogyne genotypes (effect of supergene genotypes in her own colony on the mortality of a queen: $\chi^2 = 6.52$; $p = 0.03$; $n = 56$ monogyne, 176 mixed, 60 polygyne; Fig. 3a; *post hoc* comparisons: Estimate (monogyne vs mixed) = $-0.89$; SE = 0.38; $z = -2.35$; $p = 0.03$). Queens with monogyne genotypes in colonies with monogyne workers were also less likely to die than queens with polygyne genotypes in colonies with polygyne workers (Estimate (monogyne vs polygyne) = $-1.06$; SE = 0.46; $z = -2.30$; $p = 0.03$), whereas queen mortality rates were similar for monogyne queens with polygyne workers and polygyne queens with polygyne workers (Estimate (mixed vs polygyne) = $-0.17$; SE = 0.35; $z = -0.49$; $p = 0.62$). Specifically, 37.5% and 38% of queens died in the colonies with mixed and polygyne supergene genotypes, respectively, whereas only 26% of queens died in the colonies with monogyne genotypes (Fig. 3a). The size difference between the two colonies (the difference in the number of workers at the start of the experiment) significantly predicted queen mortality, with higher death risk for queens in the smallest colony of a pair (Estimate = $-0.003$; SE = 0.001; $z = -2.22$; $p = 0.02$).

Queens were also more likely to die when facing a colony with monogyne genotypes than when facing a colony with either mixed or polygyne genotypes (effect of supergene genotypes in the second colony on queen mortality in the first colony: $\chi^2 = 14.61$; $p = 0.0006$; $n = 56$ monogyne, 176 mixed, 60 polygyne; Fig. 3b; *post hoc* comparisons: Estimate (monogyne vs mixed) = 1.28; SE = 0.36; $z = 3.50$; $p = 0.001$; Estimate (monogyne vs polygyne) = 1.64; SE = 0.48; $z = 3.39$; $p = 0.001$). Queens had similarly small probabilities of dying when facing colonies with mixed or polygyne genotypes (Estimate (mixed vs polygyne) = 0.36; SE = 0.38; $z = 0.95$; $p = 0.33$). Specifically, 57% of queens died when facing a colony with monogyne supergene genotypes, whereas 32% and 25% died when facing a colony with mixed or polygyne genotypes, respectively (Fig. 3b). When we re-ran the analysis with z-scored size differences between paired laboratory colonies, we found qualitatively similar results (effect of supergene genotypes in her own colony on the mortality of a queen: $\chi^2 = 6.52$; $p = 0.03$; effect of supergene genotypes in the second colony on queen mortality in the first

colony: $\chi^2 = 14.61$; $p = 0.0006$), as we did when we removed all random effects except for field colony id (effect of supergene genotypes in her own colony on the mortality of a queen: $\chi^2 = 6.86$; $p = 0.03$; effect of supergene genotypes in the second colony on queen mortality in the first colony: $\chi^2 = 17.99$; $p = 0.0001$; size difference among the colonies: $\chi^2 = 5.75$; $p = 0.02$).

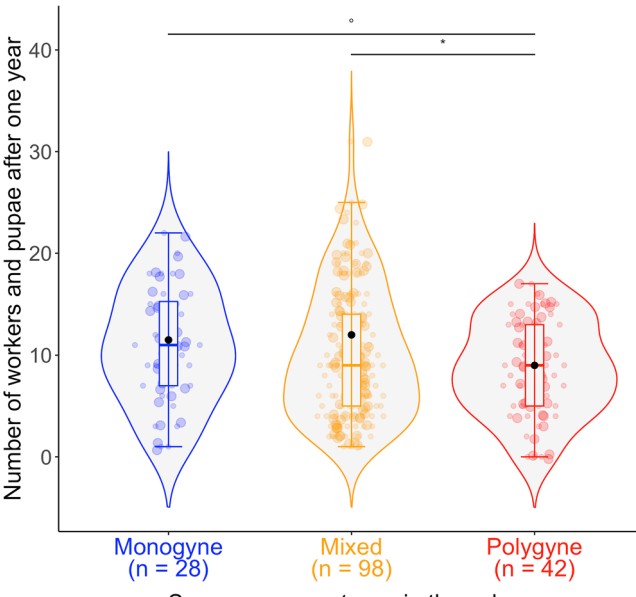

**Fig. 1 | Colonies with mixed supergene genotypes were as large as those with only monogyne genotypes, but larger than those with polygyne genotypes.** The graph shows the size of the colony after a year (the total number of workers and pupae), according to the supergene genotypes in the colony. Monogyne = *MM* queen and *MM* workers; Mixed = *MM* queen and *MP* workers; Polygyne = *MP* or *PP* queen and workers. The horizontal line marks the median colony sizes, and the boxes englobe 50% of the data. *$p < 0.05$; °$p < 0.1$. Black dots reflect fitted median values extracted from the model, per supergene genotypes in the colony.

## Discussion

In several independent ant clades, supergenes control the number of queens living in each colony, with the derived haplotypes promoting the presence of multiple queens[10–14,45]. These supergenes play a major role in determining the phenotype of mature colonies, as well as the morphology and behavior of queens, males, and workers produced by them[9,17,18,20–23,46–48]. Yet, as many physiological, behavioral, and social factors are correlated in the field, and because supergenes have indirect effects on individuals by altering their social environment, the direct genetic effect of supergenes on behavior, and the precise ontogenetic changes leading to the development of alternative forms of social organization, remained poorly understood.

In fire ants, colony social organization seems to critically depend on worker behavior towards queens and on the proportion of workers carrying the derived supergene haplotype (*b*) within polygyne colonies[20,22,23,49–51]. Here, we provide experimental evidence that an independently evolved supergene has a direct genetic effect on the worker caste in Alpine silver ants, which steers the colony social organization towards multiple queening. When workers from monogyne mothers inherited the derived supergene haplotype (*P*) from their fathers, their incipient colonies were more likely to fuse with other *P*-carrying colonies. Therefore, one *P* haplotype in workers is sufficient to switch the developmental trajectory of their social group from a single-queen colony to a multi-queen colony, overriding the supergene genotype of their mother, which promotes single-queening. The dominant *P* haplotype thus induces colony-level phenotypic changes from early stages of colony development, through a direct effect on the worker caste.

The effect of the *P* haplotype in workers indicates that *MM* queens mated with *P* males establish colonies that are likely to become multi-queened. This would explain why no mature single-queen colony of this cross has been found in the field in Alpine silver ants[12,28]. This absence was puzzling, as *MM* queens commonly mate with *P* males in the field[25]. Moreover, this cross produces offspring in the laboratory[32–34], and establishes incipient colonies in the wild[29]. Our experiment reveals that colonies with mixed supergene genotypes (*MM* queens and *MP* workers) tend to fuse with other colonies that contain unrelated queens and workers with polygyne supergene genotypes, leading to the formation of multi-queen colonies. Moreover, positive relatedness among nestmate queens[28] suggests that *P*-carrying colonies can later readopt daughter queens, leading to the formation and maintenance of multi-queen colonies.

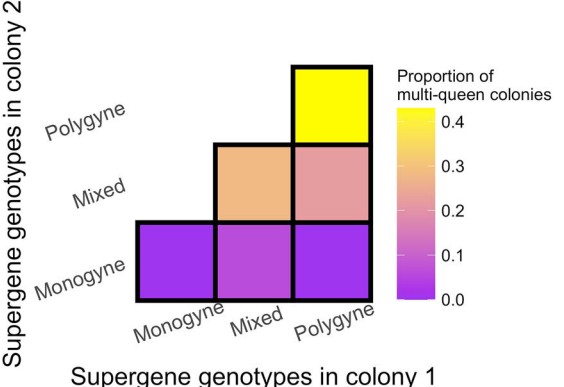

**Fig. 2 | A *P* haplotype in workers made the colony more likely to fuse with other *P*-carrying colonies, independently of the genotype of the queen.** The graphs show the proportion of replicates (i.e., pairs of colonies) that became a multi-queen colony, according to the supergene genotypes in each colony within a replicate.

Monogyne = *MM* queen and *MM* workers; Mixed = *MM* queen and *MP* workers; Polygyne = *MP* or *PP* queen and workers. The mean and 95% CI across pairs of colonies are indicated.

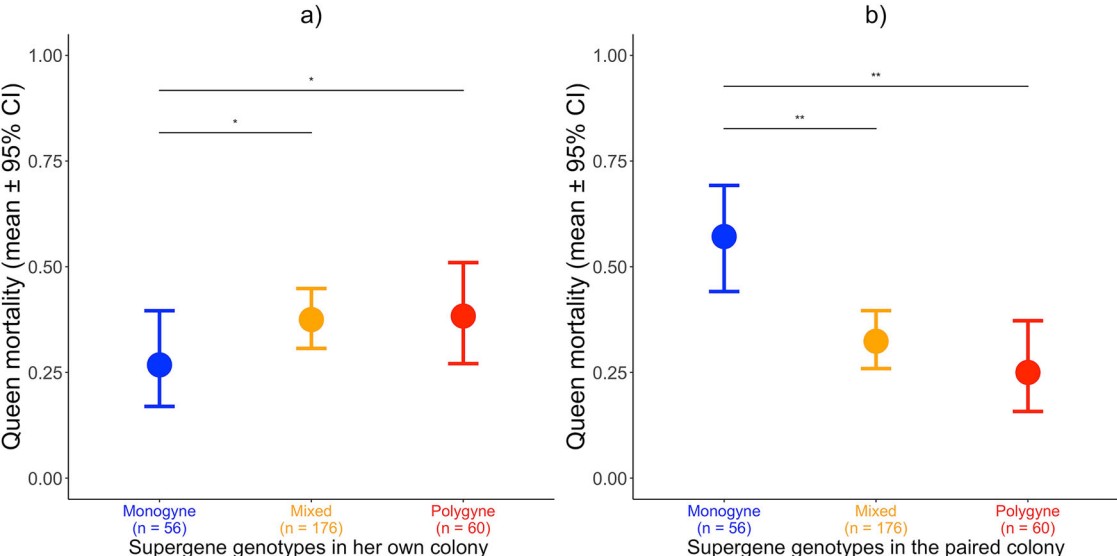

**Fig. 3 | A *P* haplotype in workers made the colony more likely to lose her own queen, and more likely to accept the other queen, independently of the genotype of the queen.** Both figures show queen mortality (the proportion of colonies that lost their queens) in incipient single-queen colonies paired with other single-queen colonies, according to the supergene genotypes in the queen's own colony (**a**), and in the colony with which her colony was paired (**b**). Monogyne = *MM* queen and *MM* workers; Mixed = *MM* queen and *MP* workers; Polygyne = *MP* or *PP* queen and workers. The graphs show the mean and 95% CI of queen mortality. Post-hoc FDR corrected *$p < 0.05$; **$p < 0.001$.

The independently evolved supergene haplotypes that induce polygyny in two ant clades, *P* (in *Formica*) and *b* (in *Solenopsis*), show convergent properties, despite a lack of homology at the gene level[12]. Both are selfish genetic elements favoring their own transmission, albeit through different mechanisms[22,31]. Both carry a severe genetic load, which causes mature polygyne colonies to contain an excess of heterozygote individuals[11,35]. Specifically, *P* recessive lethality increases the mortality of *PP* females by 50% in polygyne colonies of *F. selysi*[35], while *bb* females rarely survive at all in *S. invicta*[15]. Finally, the possibility that monogyne queens generate polygyne colonies by mating with males of polygyne origin was first suggested in the fire ant, *S. invicta*[24]. Our results show that in Alpine silver ants, colonies established by one queen with a monogyne supergene genotype (*MM*) mated to a male carrying the polygyne haplotype (*P*) tend to become multi-queen colonies, while benefitting from the high fertility and colony founding abilities of *MM* queens[18]. This cross therefore likely contributes to the spread of the polygyne social organization in the wild.

The effects of the workers' supergene on colony development may generate conflicts between sexes and social forms. This is because monogyne queens, which have the *MM* supergene genotype, are likely to have reduced fitness when they mate with males of polygyne origin (carrying the *P* haplotype), as compared to males of monogyne origin (carrying the *M* haplotype). Such a cross will produce an incipient colony with workers that are more likely to lose their queen, and that will eventually be prone to become a multi-queen colony. Although the lifetime inclusive fitness of queens is difficult to estimate, fusing with other incipient colonies or adopting additional queens are likely to reduce the fitness of the founding queen, so that *MM* queens may be selected to avoid mating with *P* males. Interestingly, although 57% of males engaging in swarms have the *P* haplotype, 80% of the *MM* queens mate with *M* males[25,28]. The mechanism underlying this strong assortative mating by social forms remains enigmatic[18,32], and may involve male-male competition or cryptic female choice.

The multi-queen phenotype of the colony was determined by the supergene genotypes of the workers, independently of the genotype of the founding queen. Our incipient laboratory colonies were very small, and the smallest colonies that became multi-queened had fewer than 10 workers, suggesting that the supergene genotype of a few workers overrules the one of the queen and changes the colony phenotype. Worker control of colony-level traits seems frequent in the social insects. For example, the resistance of honeybee colonies against brood diseases can be explained by the cumulative hygienic behavior of a few individuals[52–54]. Similarly, in ants, bees, and wasps, the secondary sex-ratio often matches the worker's (1 male to 3 females) rather than the queen's (1:1) evolutionary equilibrium[55,56]. Our results are another illustration of the power of the worker caste in determining colony phenotype.

Our experiment provides evidence for a direct genetic effect of supergenes on ant social phenotypes. Although supergenes are correlated with a range of discrete phenotypes[57–59], showing their direct effect on social phenotypes is difficult because experimental breeding is tricky and many social, morphological, and behavioral traits correlate with the supergene genotype in the field. In previous experiments, we showed that the supergene had a direct effect on the morphology and dispersal abilities of Alpine silver ant queens. Specifically, *MP* and *PP* queens produced by the same mothers and raised by the same workers within the same polygyne colonies differed in morphology and dispersal-related traits[28], which ruled out indirect genetic effects, such as social or environmental effects due to the genotype of other group members[60,61]. Here, we show that the tolerance of workers to additional queens is directly influenced by their supergene genotype, because *MP* workers (in colonies with mixed supergene genotypes) and *MM* workers (in colonies with only monogyne supergene genotypes) were both produced by *MM* queens in single-queen laboratory colonies, mated to either a *P* or a *M* male, respectively, and males do not contribute to parental care.

In our experiment, the workers' supergene genotype influenced whether colonies became multi-queened and thus whether workers tolerated extra queens and succeeded in protecting their queen. In a previous experiment, we recorded workers directly killing alien queens, by allowing only workers (and not queens) to interact with alien queens. In that previous experiment, workers from every tested colony killed queens within 48 h[26]. In the present experiment, we occasionally observed the workers attacking the queen as well, but we did not formally record behavioral interactions. The proximate mechanisms through which the *P* haplotype in workers influences colony social structure remain to be further investigated. Several non-mutually exclusive mechanisms could operate. For example, *P* could make workers more tolerant towards non-nestmates, by influencing their own colony odor, or their ability to detect the odor of non-nestmates. *P* could also affect workers' ability to detect extra queens, or their fighting abilities.

In summary, this experimental study of Alpine silver ants helps to understand how supergenes lead to alternative forms of colony social organization. It shows that the *Formica* social supergene influences colony phenotype from very early stages of colony development. Indeed, a single paternally-inherited *P* haplotype in workers suffices to tilt the colony phenotype from a single-queen social organization specified by the queen's supergene genotype to a multi-queen organization specified by the workers' supergene genotype. This dominant effect of the derived supergene haplotype on colony social structure may contribute to the spread of multi-queen colonies in the field. This spread may be offset by the tendency of monogyne queens to mate assortatively and to have faster growing colonies, which might help explain the persistence of both social forms. More generally, our controlled experiment provides direct evidence that the *P* haplotype present in workers steers colony social organization towards multiple queens in *Formica* ants.

## Data availability

Data are available from Figshare: https://doi.org/10.6084/m9.figshare.28938974[62].

## Code availability

Codes are available from Figshare: https://doi.org/10.6084/m9.figshare.28938974[62].

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

## Acknowledgements

We thank Patrick Januario Lopez, Cristophe Lakatos, Sagane Joye-Dind, Jason Buser, Consolée Aletti, Julie Guenat, and Santiago Herce Castañón for their help with field and laboratory work. This work was supported by the Swiss National Science Foundation (grant numbers 31003 A-173189 / 1 and 310030-207642 / 1).

## Author contributions

O.D.G., P.B., Ma.C. and M.C.: conceived the idea; O.D.G. and P.B.: conducted the experiments and collected the data, O.D.G. analyzed the data and created Figures, O.D.G. and M.C.: wrote the first draft of the manuscript, O.D.G., P.B., Ma.C. and M.C. contributed to the final manuscript.

## Competing interests

The authors declare no competing interests.

## Ethics

This work complied with the relevant legal requirements of the University of Lausanne and Switzerland. *F. selysi* is not an endangered species.
