## [Transparent Peer Review file · Communications Biology]

Supergene regulation of ant social organization: a P haplotype in workers shifts colony ontogeny towards multiple queens

Corresponding Author: Dr Ornela De Gasperin

Version 0:

Reviewer comments:

Reviewer #1

(Remarks to the Author)

In this study, the authors assess the biological implications of supergenes, an interesting topic in evolutionary ecology that has been well studied in a few ant genera. As the authors carefully report in their introduction, many studies have been performed on *Solenopsis* ants first and *Formica* later on, to understand how supergenes regulate social organisation of ant colonies: despite this, many aspects still need clarification and the interplay between queen and worker genotypes is an aspect that has been challenging to test. This study is novel in the sense that it attempts to disentangle the roles played by the two genotypes, placing more weight on the role of workers to explore how their genetic background alone influences colony structure. This is well written and methodologically sound paper. It is clear that the experiments described here were also used as the basis for two previous publications, well acknowledged in the manuscript. I have basically one major concern that relates to the framing of the study and its findings, and specifically the high relevance given to the behavioural component that starts in the abstract and continues in the methods, results and discussion section. There are no behavioural assays described in this study, despite the continue reference to aggressiveness and interactions among colonies. Aggression or other social behaviours in ants can be easily observed and quantified using ranking systems that include for example antennation, biting, chasing/escaping, etc...but there is no mention of any of these: did the authors record any of these interaction among workers and queens in the different combinations of genotypes? I believe that a careful quantification of behaviour – with appropriate statistical analyses – is a requirement for a study that claims to show how the genetic background of workers influence colony aggressiveness. My understanding is that the levels of aggression were indirectly inferred looking at queen survival, and low survival was correlated with high aggression. There is some truth in this of course, and the replication of combinations was very high in some of the scenarios: nevertheless, I don't believe this is enough to make claims such as "The presence of a paternally-inherited P haplotype in workers was sufficient to make the colonies less aggressive", or "The dominant behavioral effect of the P haplotype contributes to the spread of the multi-queen social organization." and "This controlled experiment provides direct evidence that the P haplotype influences worker behavior", as reported in the abstract and repeatedly claimed in many parts of the discussion. It makes more sense in my opinion to reframe the study as an investigation of the effect of a supergene in workers in the regulation of colony structure / social organisation (as also stated in the title) or in the tolerance of additional queens. In addition, I only have a few minor comments about clarity in the text, some of them pertaining what discussed above.

Line 126: how exactly did the authors control for using colonies more than once in some experiments? Also, some combinations were totally missing for some of the years and age groups: was this controlled for as well in the statistical analyses?

Line 145: in the "colony size" section, it is not clearly explained how size was estimated...Was the total number of individuals counted, both immature and adults? Both workers and sexuals? What about eggs?

Line 157: "Colony aggressiveness" is not the most appropriate title for this section, as there was no aggression assay really. It is more like a mortality check.

Fig. 2: it's not clear how queen's phenotype and phenotype of colony paired play together to affect mortality

Line 233: were any aggressive interactions recorded/observed though?

Reviewer #2

(Remarks to the Author)

It is well-known that supergene affects some ant queen behaviour, and affects queen numbers. This study examines how the supergene genotype workers affect colony phenotypes such as colony size, queen mortality and number of queens. The topic is interesting, the manuscript is well-written, and experiments and statistical analyses are also carefully carried out. It could be an important study in the field of social evolution. The amount of details provided also makes the study reproducible. However, in my opinion, the results shown do not seem to align with the claims that the authors made. For example: the manuscript discusses how the P haplotype alters worker behaviour, but it does not present any behavioural data for workers. Including such data would strengthen the study. This study faces the same issue with the results for colony aggressiveness. I could understand that less aggressive colonies might induce lower queen mortality, but lower queen mortality does not necessarily result from lower aggressiveness. Especially in this study, queen numbers seem to be a very strong predictor of queen mortality.

Besides these general comments, I wish the authors to address also the following specific points. The manuscript can be considered for publication if these points are addressed.

Line 28: Not all ants can be classified as "superorganisms." For instance, some ants, such as *Harpegnathos saltator*, possess totipotent workers. I recommend exercising caution in using the term here.

Line 37: The term "social organisation" can have multiple interpretations. To enhance clarity, consider providing a more detailed definition or examples of what you mean by "social organisation." Additionally, elaborating on the genetic mechanisms that might contribute to variations in social organisation would be helpful. The phrase "genetically determined" seems overly broad and could benefit from a more precise explanation.

Line 41: The concept of "social supergenes" may not be familiar to all readers. Including a brief explanation or background information would make this term more accessible to a general audience.

Line 73: What are the haplotypes present in queens from polygyne colonies? Please provide this information for completeness.

Lines 78–81: The suggestion that workers with the P haplotype may accept other queens appears to be a hypothesis introduced in the manuscript. However, the rationale behind this hypothesis is not clearly explained. As you mentioned earlier, the effects of supergenes on worker behaviour are not well-studied. Consider providing a clearer justification for this hypothesis.

Line 82: Specify the types of worker behaviour and aspects of social organisation you are investigating. This will help readers understand the scope and focus of your study.

Line 148: Were the initial colony sizes of all incipient colonies equal before the experiment? If not, you need to account for differences in initial colony size in your model. Additionally, the sample size disparity between treatments is substantial, which might inflate the statistical differences between treatments. For example, a larger sample size in the mixed genotype treatment increases the likelihood of detecting significant differences by chance. I recommend conducting a bootstrapping procedure: randomly sample a subset of data from the mixed treatment to match the sample sizes of the monogyne and polygyne treatments, then repeat the analysis multiple times to evaluate the consistency of your results.

Line 169: If you include only stock colony ID as a random effect, does it influence your model's outcomes? For example, some stock colonies might exhibit higher aggression levels than others. Please verify and report this.

Line 211: Did all incipient colonies have the same colony size? If not, this needs to be controlled in the model. I also suggest standardizing colony size, for example, using a z-score, to see if this affects the results.

Line 213: The figure is visually appealing. Is it generated from the observed counts directly, or from your model's fitted values (predictions)? Ideally, you could plot the real effects predicted by your models and include the original observational data points. This approach would allow for a better assessment of the strength of the treatment effects.

Lines 220–248: The claims about colony aggressiveness in this section are based solely on queen mortality. I am not fully convinced that decreased queen mortality equates to reduced colony aggressiveness. At line 244, the authors define queen mortality as the proportion of colonies that lost their queens. In multi-queen colonies, do you count a colony as having lost its queen when only one queen has died, or does it require all queens to be dead? If it is the latter, the observed lower queen mortality in multi-queen colonies compared to single-queen colonies might be a systematic consequence of this definition.

Reviewer #3

(Remarks to the Author)

In this study, the authors investigate how the presence of a social supergene in either queens and/or workers, influences the probabilities of incipient colony fusion and/or queen death, thus testing the direct genetic effect of the supergene. They established young colonies composed of single queens of phenotype M or P, mated to a single male of either the same or alternate social genotype/phenotype (M or P), yielding colonies of type M/M, M/P, or P/P (queen phenotype/worker phenotype). When nests were between 1-3 years old, they censused colonies and gave each one access to a second colony with either the same or an alternate social structure, in a full factorial design resulting in 6 pairing types. They then tracked queen survival and colony fusion outcomes over a 2-month period. At one year, M/M colonies were largest, and P/P smallest, with M/P intermediate in size. The probability of colony fusion was lowest for M/M colonies paired with other M/M colonies, or when M/M was paired with P/P; highest when P/P colonies were paired together; and intermediate for the three combinations that included at least one colony with mixed or mismatched queen and worker phenotypes (M/P). With regards to individual queen survival, monogyne queens with monogyne workers (M/M) were more likely to survive than either monogynous queens with polygynous workers (M/P) or polygynous queens with polygynous workers (P/P), with queen survival rates being similar for the latter two queen contexts. Related to this, a focal queen's survival was lowest when any of the three colony types was up against an M/M colony, and equivocal when any of the types were paired with either an M/P or P/P colony. The authors conclude that this means that M queens mating with P males will produce colonies that are more likely to fuse, which could facilitate spread of multi-queen colonies whenever this mating outcome occurs. They further assert that this experiment provides good empirical evidence for a direct genetic effect of supergenes on ant social phenotypes, as M queens mated to P males can produce both P and M workers, and the resulting colonies display the increased probability of fusion, in spite of the queen's genotype.

This paper is important for demonstrating with a good experimental design that cooperative worker genotypes directly affect the propensity for colonies to fuse and tolerate the presence of multiple queens in this species, regardless of whether the queen shares this genotype, facilitating the spread of cooperation/polygyny. It is also exciting for showing a direct genetic effect of the social supergene in this species, a feat which is challenging in social insects due to the technical challenge of controlling queen matings. However, at present some context is missing, clarity is lacking, and there are some leaps of logic that need to be addressed to better highlight the findings and support the authors' assertions.

Introduction:

Provide a few more specific background points on the natural history of the focal *Formica* species, to provide context for the experiments carried out, to the extent the following are known or unknown:

What is known about the geographic distribution of polygyny in this species?

To what extent, specifically, do multi-queen colonies start via primary polygyny versus secondary polygyny?

Is this social supergene occurring in an introduced range, as for *Solenopsis*, or in the species' historic range?

Do queens of this *Formica* species typically mate multiply, or do they generally mate once, as in this experiment?

Lines 76-78: Clarify. When you say "that has never been observed in mature colonies," do you mean to indicate that mature monogynous colonies containing MM queens with P offspring have never been observed? Or do you mean that either mature monogynous or polygynous colonies containing MM queens with P offspring have never been observed? Or is this unknown, and one of the main points of carrying out the current experiments?

Related to the need for clarification, the Introduction would benefit from more clearly stated hypotheses and predictions, particularly because of the complexity of the treatments. What does it mean that "supergenes may indirectly influence the behavior of individuals by altering their social environment"? (lines 87-88) Just put what it means in concrete terms as related to the experiment design (e.g. what P workers would do in a colony with an M mother). How would an indirect influence actually work and function differently from what was found in this experiment? In other words, what would the alternate predictions be, and how does this experiment actually distinguish between these two mechanisms? It might be most suitable to outline predictions by modifying Table 1, or this could be carried out directly in the text.

Methods:

Lines 131-135: If queens were paint-marked, in cases where only one of the two queens survived, was the queen's identity noted? What is meant by the reference to reference 26 - "following (26)"? Spatial information does not appear to be used in the eventual analyses and results for this set of experiments.

Line 111: How many field colonies supplied queens for the experiments, overall?

Line 121: Replace "all" with "each"

Line 123: How were food and water provided in this setup? Do these colonies maintain food stores or other resources that are guarded by workers?

Lines 127-128: Given that there was an a priori expectation that colony growth rate would depend on the specific queen-worker genotype combination (see also lines 146-148), how could it work to pair same-age colonies to minimize differences in brood sizes between pairs of colonies? How much bigger are colonies by year 2, for example?

Results:

It may be less confusing to present colony fusion outcomes first, and then present queen outcomes second, since the fusion outcomes more clearly map onto the initial treatment groups.

The section on "Colony aggressiveness" is generally confusing, most likely because it does not easily map from the

experiment pairings that were created, or any specific predictions. Also, no actual measures of aggression are included anywhere in the Methods or Results, just the context dependence of queen survival outcomes. Consider reframing this section around the larger question of the extent to which queen survival depended on a match/mismatch with her own workers, versus the identity of the workers in the other colony.

Lines 211-212: To what extent does this result depend on whether pupae are included or excluded from the comparison? It would be useful to include mean sizes directly in the written text in addition to depicting this information in Figure 1. Also, even though no statistical comparison is made, as noted at lines 127-128 above, it would be useful to know how colony sizes after year 1 compare to sizes after years 2 and 3, to provide context for the differences observed in year 1, and a sense as to whether size mismatches may have contributed to colony fusion outcomes.

Line 221: Were any efforts made whatsoever to track worker mortality and colony growth over the 2-month observation period? Was there any direct or anecdotal evidence to suggest that queens were killed by workers, versus by the other queen? Was there any evidence that any workers engaged in behaviors to actively protect their queen?

Lines 228-229: Awkward. Rephrase to: Mortality rates were similar for monogyne queens with polygyne workers as compared to polygyne queens with polygyne workers.

Lines 263-264: Rephrase this: When workers carried a P haplotype, the probability that a replicate would become a multi-queen colony was similar regardless of whether the queen was monogyne or polygyne.

Discussion: Mostly minor clarifications suggested.

Line 292: suggest rephrasing from "has a direct genetic effect on worker behavior" to "has a direct genetic effect determining worker behavior" for clarity

Lines 314-315: Do supergenes also consistently cause polygyne colonies to grow more slowly? (further evidence of the genetic load, at least for Formica if not Solenopsis)

Line 336: Give a concrete number here.

Lines 347-351: Logic here is unclear. Break up the sentence and clarify.

Lines 351-353: What evidence exists that there are both MP and MM workers present? In the Introduction you state that "brood from MP queens that lacks the P haplotype fails to develop." (Line 75).

Lines 355-360: These assertions would be strengthened by even just a small amount of anecdotal evidence from the current experiment (e.g. evidence of worker mortality, see comment at Line 221).

Line 363: suggest "supergene directly influences worker behavior..."

Figures and Tables:

Table 1. Simplify the text describing the colony types to improve legibility. E.g. the title of the first column could be "Colony 1 supergene genotypes" and the first entry shortened to "Mixed (MM queen, MP workers)"

Figure 1. Suggest revising the y-axis to "Number of workers plus pupae after one year" to indicate measurement unit.

Supplementary material:

Can the authors provide metadata to accompany the supplied dataset and analysis? This would aid with interpreting variables, especially given the occurrence of duplicate variable names.

Given the complexity of the analyses, additional commentary to connect data analysis scripts to specific results would be helpful.

Version 1:

Reviewer comments:

Reviewer #1

(Remarks to the Author)

The authors have performed an excellent job in addressing all comments raised by the three reviewers systematically. This has improved the manuscript's focus and clarity significantly. One minor aspect that I am still unsure of is the repeated use of the same colonies for different experiments (or lack thereof). In the rebuttal letter, at page 2, the authors say "For colony size, we included each colony only once in the analyses, when they were one year old." However, in the revised version of the manuscript, the authors say in the methods: "Out of 227 single-queen laboratory colonies, 167 were used only once, and 60 were used more than once. In the statistical analyses, we controlled for having used some laboratory colonies more than once, and for having queens emerging from the same field colonies..." I am therefore confused on what strategies was

adopted and what has been accounted for in the statistical analyses.

Reviewer #2

(Remarks to the Author)

The authors have adequately addressed all the concerns I previously raised. I recommend the manuscript for publication.

Reviewer #3

(Remarks to the Author)

I commend the authors for thoughtfully and thoroughly responding to points raised in the prior review, especially with regards to transparency in data analysis and data sharing. Overall, the relevant background on *F. selysi* is also now much more clearly explained. There are still some minor points of clarification, mostly in the Introduction and Results, that would further improve understanding of the work that was carried out, and of the major findings.

Lines 61-64: Couldn't it also be the case that queens of polygyne origin occasionally mate with males of monogyne origin? But that in field colonies, it is difficult/impossible to distinguish where there is or isn't a mismatch, so as to determine the consequences of either a match or mismatch? (to get at your main point in the paragraph at lines 65-70). I think you do address this more clearly in the paragraph at lines 71-83, so you may want to revise this paragraph accordingly if I have misinterpreted the intention here.

Line 67: To clarify this important point, I'd suggest saying "because queens and workers with each of the alternative supergene genotypes typically develop in separate colonies that differ in social organization and many associated phenotypic traits..."

Lines 77-80: Modify to "All mature colonies typically contain..." so this does not contradict other sections of the Introduction.

Line 159: suggest revising to , "...and eggs and larvae were not counted."

Line 175: suggest revising to "We compared the size of colonies (workers plus pupae) with monogyne, polygyne, and "mixed"..."

Line 303: revise to "Overall, queens in colonies containing workers with the paternally-inherited P haplotype were more likely to die...". Then at line 304, "Specifically, queens..."

Line 309: For clarity, revise to "...less likely to die than polygyne queens in colonies with only polygyne workers"

Figure 3: Is it appropriate to have the points joined together with gray lines?

Line 435: Might be useful to point out that this spread may potentially be offset by the tendency of monogyne queens to mate assortatively, and/or have colonies that grow more quickly. (to explain the overall persistence of both social forms across the species range)

Version 2:

Reviewer comments:

Reviewer #1

(Remarks to the Author)

I am happy with the answers provided by the authors in the last set of revisions.

Reviewer #1 (Remarks to the Author):

In this study, the authors assess the biological implications of supergenes, an interesting topic in evolutionary ecology that has been well studied in a few ant genera. As the authors carefully report in their introduction, many studies have been performed on *Solenopsis* ants first and *Formica* later on, to understand how supergenes regulate social organisation of ant colonies: despite this, many aspects still need clarification and the interplay between queen and worker genotypes is an aspect that has been challenging to test. This study is novel in the sense that it attempts to disentangle the roles played by the two genotypes, placing more weight on the role of workers to explore how their genetic background alone influences colony structure.

This is well written and methodologically sound paper. It is clear that the experiments described here were also used as the basis for two previous publications, well acknowledged in the manuscript.

Thank you for your kind comments.

I have basically one major concern that relates to the framing of the study and its findings, and specifically the high relevance given to the behavioural component that starts in the abstract and continues in the methods, results and discussion section. There are no behavioural assays described in this study, despite the continue reference to aggressiveness and interactions among colonies. Aggression or other social behaviours in ants can be easily observed and quantified using ranking systems that include for example antennation, biting, chasing/escaping, etc...but there is no mention of any of these: did the authors record any of these interaction among workers and queens in the different combinations of genotypes? I believe that a careful quantification of behaviour – with appropriate statistical analyses – is a requirement for a study that claims to show how the genetic background of workers influence colony aggressiveness.

We agree that the framing of the initial submission was not fully aligned with the data, as we did not conduct behavioral observations in this experiment. To address this shortcoming, we have reworded the abstract, introduction, and discussion. We now avoid speculative conclusions about aggression and focus on the regulation of social structure. In addition to changing the topic of the revised manuscript away from aggression throughout the manuscript, we now mention in the discussion that worker aggression towards queens was observed in previous experiments (lines 417-425).

My understanding is that the levels of aggression were indirectly inferred looking at queen survival, and low survival was correlated with high aggression. There is some truth in this of course, and the replication of combinations was very high in some of the scenarios: nevertheless, I don't believe this is enough to make claims such as "The presence of a paternally-inherited P haplotype in workers was sufficient to make the colonies less aggressive", or "The dominant behavioral effect of the P

haplotype contributes to the spread of the multi-queen social organization.” and “This controlled experiment provides direct evidence that the P haplotype influences worker behavior”, as reported in the abstract and repeatedly claimed in many parts of the discussion. It makes more sense in my opinion to reframe the study as an investigation of the effect of a supergene in workers in the regulation of colony structure / social organisation (as also stated in the title) or in the tolerance of additional queens.

Thank you for this suggestion, we have changed all these phrases accordingly.

In addition, I only have a few minor comments about clarity in the text, some of them pertaining what discussed above.

Line 126: how exactly did the authors control for using colonies more than once in some experiments? Also, some combinations were totally missing for some of the years and age groups: was this controlled for as well in the statistical analyses?

We apologize for not having offered enough clarity in the first version. We have now improved the description of the statistical analyses in the methods. In detail:

- For colony size, we included each colony only once in the analyses, when they were one year old. We included as random effects the colony of origin of the queen (from the field), nested within the year of the experiment (because queens collected from the same field colony within the same year are likely to be more similar to one another; specified in line 181-184). Note that as all laboratory colonies were one year old, the year reflects the year of collection from the field.
- For queen death, we included as random effects the replicate nested within the experimental year (because we were modelling the death of two queens within a replicate (1|experimental_year/replicate), and because specific conditions of a year may explain variance. We included a second random effect, which was the queen's id (because some queens survived in a year and were reused in following years), nested within the colony of origin of the queen from the field, because queens collected from the same field colony are likely to be more similar to one another. Random effects were thus specified in lines 237-242: (1| experimental_year/replicate) + (1|year_field_collection/Field_colony /Queen_code)
- For multi-queen colony formation, we did not control for the year, neither as random nor fixed effect, because the year was highly correlated with the size of the colonies (i.e., the number of workers), which is likely more biologically relevant than the year. We did not include the queen colony of origin from the field as a random effect, because Firth's penalized-likelihood logistic regression does not allow to include random effects (note that the field colony of origin explained zero variance in all other models, see R markdown files

included as ESM, for instance, analyses of queen mortality suggested by reviewer 2, with field colony id as single random effect).

Line 145: in the “colony size” section, it is not clearly explained how size was estimated...Was the total number of individuals counted, both immature and adults? Both workers and sexuals? What about eggs?

Two weeks before the start of the experiment, we counted the total number of workers and pupae in the colony. Colonies did not produce sexual individuals, and we did not count eggs. We now specify this in the manuscript (lines 157-160). We returned to our lab datasheets to separate pupae from workers, and we now specify that we controlled for colony size using worker number (previously, we used worker + pupae) (please note that our results are qualitatively similar).

Line 157: “Colony aggressiveness” is not the most appropriate title for this section, as there was no aggression assay really. It is more like a mortality check. Thank you for this suggestion, we changed the subtitle to ‘Queen mortality’.

Fig. 2: it’s not clear how queen’s phenotype and phenotype of colony paired play together to affect mortality

We have better explained our experimental design, to make clear that queen mortality depended on the supergenes of workers in pairs of incipient single-queen colonies (lines 230-233). Please note that both monogyne and mixed colonies host one single MM queen (originating from single-queen colonies), and that polygyne colonies host a single PP or MP queen (originating from multi-queen colonies).

Line 233: were any aggressive interactions recorded/observed though?

No, we did not perform behavioral observations in this experiment. We have been careful to give less weight to behavior throughout the revised manuscript, and we specify that we have frequently observed workers killing queens in this and in previous experiments (lines 419-423).

Reviewer #2 (Remarks to the Author):

It is well-known that supergene affects some ant queen behaviour, and affects queen numbers. This study examines how the supergene genotype workers affect colony phenotypes such as colony size, queen mortality and number of queens. The topic is interesting, the manuscript is well-written, and experiments and statistical analyses are also carefully carried out. It could be an important study in the field of social evolution. The amount of details provided also makes the study reproducible.

Thank you for these kind comments.

However, in my opinion, the results shown do not seem to align with the claims that the authors made. for example: the manuscript discusses how the P haplotype alters worker behaviour, but it does not present any behavioural data for workers. Including such data would strengthen the study.

We appreciate this comment, and, as we did not evaluate behaviour, we have rephrased the manuscript to give less weight to the behaviour of the workers, throughout the manuscript. We have also clarified the experimental design and the logic linking data and conclusion.

This study faces the same issue with the results for colony aggressiveness. I could understand that less aggressive colonies might induce lower queen mortality, but lower queen mortality does not necessarily result from lower aggressiveness. especially in this study, queen numbers seem to be a very strong predictor of queen mortality.

Given the lack of direct observation, we avoided the wording "colony aggressiveness" in the revised manuscript. However, we also believe there is a misunderstanding, as all colonies had a single queen. In our experiment, the term "polygyne" reflects the genotype of the individuals (*MP* or *PP*), and not the number of queens in the incipient colonies. Therefore, queen number cannot underlie queen mortality, as this factor was controlled for. Furthermore, in figure 2, monogyne and mixed colonies had similar queens (a single queen, with *MM* supergene genotype). We thus expect queen mortality to reflect the genotype of the workers (*MM* supergene genotype in monogyne colonies, and *MP* supergene genotype in mixed or polygyne colonies). We have clarified the experimental design and question being tested in the last paragraph of the introduction, and in figure 2 legend.

Besides these general comments, i wish the authors to address also the following specific points. The manuscript can be considered for publication if these points are addressed.

Line 28: Not all ants can be classified as "superorganisms." For instance, some ants, such as *Harpegnathos saltator*, possess totipotent workers. I recommend exercising caution in using the term here.

Thank you for this comment, we have now added 'most ant species' (line 30).

Line 37: The term "social organisation" can have multiple interpretations. To enhance clarity, consider providing a more detailed definition or examples of what you mean by "social organisation."

We have opted for a more precise definition in the abstract (lines 12-15) and in the introduction (lines 40, 74, 96, 109). We now use "social structure" instead of "social organization" in several other key sentences (lines 424).

Additionally, elaborating on the genetic mechanisms that might contribute to variations in social organisation would be helpful. The phrase "genetically determined" seems overly broad and could benefit from a more precise explanation. We have extended this paragraph as suggested (lines 38-43).

Line 41: The concept of "social supergenes" may not be familiar to all readers. Including a brief explanation or background information would make this term more accessible to a general audience.

We have changed this phrase to add clarity (lines 42-43).

Line 73: What are the haplotypes present in queens from polygyne colonies? Please provide this information for completeness.

We added with *MP* or *PP* supergene genotypes, accordingly (lines 79-80).

Lines 78–81: The suggestion that workers with the P haplotype may accept other queens appears to be a hypothesis introduced in the manuscript. However, the rationale behind this hypothesis is not clearly explained. As you mentioned earlier, the effects of supergenes on worker behaviour are not well-studied. Consider providing a clearer justification for this hypothesis.

We have added information in this paragraph to better explain the logic. In essence, *P*-carrying workers in multi-queen colonies tolerate extra queens, while *MM* workers keep only one queen (lines 80-83).

Line 82: Specify the types of worker behaviour and aspects of social organisation you are investigating. This will help readers understand the scope and focus of your study.

Thank you for this suggestion. We have rephrased this section (lines 96-105).

Line 148: Were the initial colony sizes of all incipient colonies equal before the experiment? If not, you need to account for differences in initial colony size in your model.

Thank you for this suggestion.

We did not manipulate colony size at the beginning of the experiment. We counted the number of workers and pupae.

Colonies were paired randomly to the size of the colonies. We now specify that: We paired same-age colonies (except for one pair) to minimize differences in brood size

between colonies, but matching was done randomly with respect to worker number (lines 151-154).

For queen mortality, we included the size difference between both colonies in the statistical model, as we expected the largest colony to be more likely to kill the other queen. This is the pattern that we find, the queen has a higher risk to die if her colony is smaller than the other queen's colony. We have added this in the manuscript (lines 235-237).

For the propensity to become multi-queened, we included in the model the overall size of the two colonies (the total number of workers in the two colonies paired in a replicate). This is because, as colonies were paired within a year, the size of colony 1 was highly correlated with the size of colony 2 ($\rho = 0.75$, $p\text{-value} < 10^{-15}$), and we do not expect that a small difference will affect the probability of becoming multi-queened. We now report that the overall number of workers did not affect the outcome (lines 285-286).

Additionally, the sample size disparity between treatments is substantial, which might inflate the statistical differences between treatments. For example, a larger sample size in the mixed genotype treatment increases the likelihood of detecting significant differences by chance. I recommend conducting a bootstrapping procedure: randomly sample a subset of data from the mixed treatment to match the sample sizes of the monogyne and polygyne treatments, then repeat the analysis multiple times to evaluate the consistency of your results.

Thank you for this comment. We have taken a simple approach of rerunning the analysis on the three treatments with large sample sizes, mixed-monogyne, mixed-mixed and mixed-polygyne. This confirmed that the probability to become multi-queened was lower for the mixed-monogyne replicates than for the mixed-mixed and mixed-polygyne replicates. We also draw the bootstrapped distributions of these three treatments with large sample sizes (bootstrapped each 1000000 times, with replacement, and resampling 12 replicates). We included these bootstrapped distributions in the new ESM. The bootstrapped distributions confirm the general pattern, but please note that they are not directly comparable to the three other treatments that had lower sample sizes, as the proportion of replicates that became multi-queened was 0 in two of these three treatments (monogyne-monogyne and monogyne-polygyne).

Line 169: If you include only stock colony ID as a random effect, does it influence your model's outcomes? For example, some stock colonies might exhibit higher aggression levels than others. Please verify and report this.

Thank you for this suggestion. We have re-ran the model including only colony ID as a random effect, and the results are qualitatively similar. We report this in the manuscript (lines 326-329). Note that colony ID explained zero variance. We also re-ran the model including only laboratory colony ID as random effect, and none of

the results changed. This is supplied in the R Markdown files and output, all of which will be uploaded into Dryad.

Line 211: Did all incipient colonies have the same colony size? If not, this needs to be controlled in the model. I also suggest standardizing colony size, for example, using a z-score, to see if this affects the results.

Thank you for this suggestion. We have re-ran the model with a z-score for colony size and reported it (lines 322-326).

Line 213: The figure is visually appealing. Is it generated from the observed counts directly, or from your model's fitted values (predictions)? Ideally, you could plot the real effects predicted by your models and include the original observational data points. This approach would allow for a better assessment of the strength of the treatment effects.

Thank you for this suggestion. The graph now shows the observed values and the black dots represent the median fitted values per group, as suggested.

Lines 220–248: The claims about colony aggressiveness in this section are based solely on queen mortality. I am not fully convinced that decreased queen mortality equates to reduced colony aggressiveness.

Thank you for this suggestion. We have changed the subtitle to 'Queen mortality'.

At line 244, the authors define queen mortality as the proportion of colonies that lost their queens. In multi-queen colonies, do you count a colony as having lost its queen when only one queen has died, or does it require all queens to be dead? If it is the latter, the observed lower queen mortality in multi-queen colonies compared to single-queen colonies might be a systematic consequence of this definition. All colonies had a single queen in this study, and the terms "polygyne" and "mixed" are referring to their supergene genotypes. We have better explained the design in the revised manuscript (last paragraph of the introduction and Figure 2 legend).

Reviewer #3 (Remarks to the Author):

In this study, the authors investigate how the presence of a social supergene in either queens and/or workers, influences the probabilities of incipient colony fusion and/or queen death, thus testing the direct genetic effect of the supergene. They established young colonies composed of single queens of phenotype M or P, mated to a single male of either the same or alternate social genotype/phenotype (M or P), yielding colonies of type M/M, M/P, or P/P (queen phenotype/worker phenotype). When nests were between 1-3 years old, they censused colonies and gave each one access to a second colony with either the same or an alternate social structure, in a full factorial design resulting in 6 pairing types. They then tracked queen survival and colony fusion outcomes over a 2-month period. At one year, M/M colonies were largest, and P/P smallest, with M/P intermediate in size. The probability of colony fusion was lowest for M/M colonies paired with other M/M colonies, or when M/M was paired with P/P; highest when P/P colonies were paired together; and intermediate for the three combinations that included at least one colony with mixed or mismatched queen and worker phenotypes (M/P). With regards to individual queen survival, monogyne queens with monogyne workers (M/M) were more likely to survive than either monogynous queens with polygynous workers (M/P) or polygynous queens with polygynous workers (P/P), with queen survival rates being similar for the latter two queen contexts. Related to this, a focal queen's survival was lowest when any of the three colony types was up against an M/M colony, and equivocal when any of the types were paired with either an M/P or P/P colony. The authors conclude that this means that M queens mating with P males will produce colonies that are more likely to fuse, which could facilitate spread of multi-queen colonies whenever this mating outcome occurs. They further assert that this experiment provides good empirical evidence for a direct genetic effect of supergenes on ant social phenotypes, as M queens mated to P males can produce both P and M workers, and the resulting colonies display the increased probability of fusion, in spite of the queen's genotype.

Thank you for this detailed and accurate summary of our results.

This paper is important for demonstrating with a good experimental design that cooperative worker genotypes directly affect the propensity for colonies to fuse and tolerate the presence of multiple queens in this species, regardless of whether the queen shares this genotype, facilitating the spread of cooperation/polygyny. It is also exciting for showing a direct genetic effect of the social supergene in this species, a feat which is challenging in social insects due to the technical challenge of controlling queen matings. However, at present some context is missing, clarity is lacking, and there are some leaps of logic that need to be addressed to better highlight the findings and support the authors' assertions.

Thanks for these comments. We have added context and clarified the specific points you raised, as detailed below.

Introduction:

Provide a few more specific background points on the natural history of the focal

Formica species, to provide context for the experiments carried out, to the extent the following are known or unknown:

What is known about the geographic distribution of polygyny in this species?

The species has monogyne and polygyne colonies throughout the Alpine range (Purcell et al. Mol. Ecol. 2015; Fontcuberta et al. 2022). Added on lines 72-73.

To what extent, specifically, do multi-queen colonies start via primary polygyny versus secondary polygyny?

Field data and laboratory experiments indicate rare events of joint colony founding by multiple queens and fusion of small colonies (Blacher et al., BES 2021; De Gasperin, Behav. Ecol, 2021). Genetic data indicate that nestmate queens are as related as workers, indicating that acceptance of nestmate queens is common in multi-queen colonies (Avril et al. Mol Ecol 2019). Now mentioned on lines 80-83.

Is this social supergene occurring in an introduced range, as for Solenopsis, or in the species' historic range?

The supergene occurs throughout the native range of the species. *Formica selysi* is not invasive. Added lines 72-76.

Do queens of this Formica species typically mate multiply, or do they generally mate once, as in this experiment?

Formica selysi queens usually mate once in the field, especially in single-queen colonies (Chapuisat et al. Evolution 2004, Purcell et al. Evolution 2013, Avril et al. Mol Ecol 2019). Now mentioned on line 76.

Lines 76-78: Clarify. When you say “that has never been observed in mature colonies,” do you mean to indicate that mature monogynous colonies containing MM queens with P offspring have never been observed? Or do you mean that either mature monogynous or polygynous colonies containing MM queens with P offspring have never been observed? Or is this unknown, and one of the main points of carrying out the current experiments?

Thank you for this comment. We have clarified that we have never detected an MM queen with MP workers (i.e., mated to a P male) in mature single-queen or multi-queen colonies (Chapuisat et al. Evolution 2004, Purcell et al. Evolution 2013, Avril et al. Mol Ecol 2019) (lines 89-97). Please note that MM queens mating with P males and heading incipient colonies have been documented in the field (Blacher et al. BES 2021) (lines 86-89).

Related to the need for clarification, the Introduction would benefit from more clearly stated hypotheses and predictions, particularly because of the complexity of the treatments. What does it mean that “supergenes may indirectly influence the behavior of individuals by altering their social environment”? (lines 87-88)

We have better explained potential indirect genetic effects, and how our experiment control for them (lines 100-105).

Just put what it means in concrete terms as related to the experiment design (e.g. what P workers would do in a colony with an M mother). How would an indirect influence actually work and function differently from what was found in this experiment? In other words, what would the alternate predictions be, and how does this experiment actually distinguish between these two mechanisms? It might be most suitable to outline predictions by modifying Table 1, or this could be carried out directly in the text.

We have reworded the paragraph to better explain the logic of the experiment and the difference between direct and indirect genetic effects (lines 100-105).

Methods:

Lines 131-135: If queens were paint-marked, in cases where only one of the two queens survived, was the queen's identity noted? What is meant by the reference to reference 26 - "following (26)"? Spatial information does not appear to be used in the eventual analyses and results for this set of experiments.

Yes, we noted the queen's identity. We have now made clear that queens were paint-marked with either green or white (randomly chosen color) to identify which colony they were from, and that an observer blind to the color code recorded which queen survived, line 164.

We have also explained how we used spatial location of queens, which was relevant because we considered that the colonies merged if both queens were alive and living in the same nest (lines 163-166).

Line 111: How many field colonies supplied queens for the experiments, overall? We used 66 unique colonies, 43 monogyne and 23 polygyne unique colonies (lines 131-133).

Line 121: Replace "all" with "each"
Changed, thank you (line 139).

Line 123: How were food and water provided in this setup? Do these colonies maintain food stores or other resources that are guarded by workers? We have now given details, thank you. Food and water were provided *ad libitum*. Water was provided inside colonies, and food inside the foraging arena (141-146).

Lines 127-128: Given that there was an a priori expectation that colony growth rate would depend on the specific queen-worker genotype combination (see also lines 146-148), how could it work to pair same-age colonies to minimize differences in brood sizes between pairs of colonies? How much bigger are colonies by year 2, for example?

We had no strong prior expectations on colony growth depending on queen and worker genotypes, but there were for colony age. Mean worker number after a year was 6, mean worker number after two years was 199 (lines 153-154).

Results:

It may be less confusing to present colony fusion outcomes first, and then present queen outcomes second, since the fusion outcomes more clearly map onto the initial treatment groups.

We have changed the order, accordingly, thank you.

The section on “Colony aggressiveness” is generally confusing, most likely because it does not easily map from the experiment pairings that were created, or any specific predictions. Also, no actual measures of aggression are included anywhere in the Methods or Results, just the context dependence of queen survival outcomes. Consider reframing this section around the larger question of the extent to which queen survival depended on a match/mismatch with her own workers, versus the identity of the workers in the other colony.

Thank you for this suggestion. We changed the subtitle to ‘Queen mortality’. We have also reframed this section along the lines you suggested and better explained the outcome of this experiment.

Lines 211-212: To what extent does this result depend on whether pupae are included or excluded from the comparison? It would be useful to include mean sizes directly in the written text in addition to depicting this information in Figure 1. Also, even though no statistical comparison is made, as noted at lines 127-128 above, it would be useful to know how colony sizes after year 1 compare to sizes after years 2 and 3, to provide context for the differences observed in year 1, and a sense as to whether size mismatches may have contributed to colony fusion outcomes.

Thank you for this comment.

We have gone through our laboratory notes, to extract specifically worker and pupae numbers. In doing so we also found more replicates. We now use worker number as explanatory variable in our models, rather than colony size (pupae + workers, as we did previously).

Please note that all our results are qualitatively similar.

Line 221: Were any efforts made whatsoever to track worker mortality and colony growth over the 2-month observation period? Was there any direct or anecdotal evidence to suggest that queens were killed by workers, versus by the other queen? Was there any evidence that any workers engaged in behaviors to actively protect their queen?

We now mention that workers killed queens in a previous experiment where lone alien queens were introduced into a colony (De Gasperin et al. Beh Ecol 2021) (lines 419-423).

Lines 228-229: Awkward. Rephrase to: Mortality rates were similar for monogyne queens with polygyne workers as compared to polygyne queens with polygyne

workers.

Changed, thank you (line 311).

Lines 263-264: Rephrase this: When workers carried a P haplotype, the probability that a replicate would become a multi-queen colony was similar regardless of whether the queen was monogyne or polygyne.

Changed, thank you (lines 276-280).

Discussion: Mostly minor clarifications suggested.

Line 292: suggest rephrasing from “has a direct genetic effect on worker behavior” to “has a direct genetic effect determining worker behavior” for clarity

Changed, thank you (we removed most behavior terms).

Lines 314-315: Do supergenes also consistently cause polygyne colonies to grow more slowly? (further evidence of the genetic load, at least for Formica if not Solenopsis)

We have added information on the magnitude of load (lines 370-374).

Line 336: Give a concrete number here.

Changed, thank you (line 395-396).

Lines 347-351: Logic here is unclear. Break up the sentence and clarify.

Changed, thank you (lines 407-411).

Lines 351-353: What evidence exists that there are both MP and MM workers present? In the Introduction you state that “brood from MP queens that lacks the P haplotype fails to develop.” (Line 75).

Changed for clarity, thank you (lines 411-416).

What we meant was that colonies with monogyne genotypes were produced by single MM queens mated to M males, with MM workers, while colonies with mixed genotypes were produced by single MM queens mated to P males, and thus had MP workers.

Because males are haploid, we are certain that colonies with monogyne genotypes had a single MM queen and MM workers, and that colonies with mixed genotypes had a single MM queen and MP workers. We clarified this in the text.

Lines 355-360: These assertions would be strengthened by even just a small amount of anecdotal evidence from the current experiment (e.g. evidence of worker mortality, see comment at Line 221).

We have shown in a previous experiment that workers kill queens. We have added this in the MS (lines 419-423).

Line 363: suggest “supergene directly influences worker behavior...”
Changed, thank you.

Figures and Tables:
Table 1. Simplify the text describing the colony types to improve legibility. E.g. the title of the first column could be “Colony 1 supergene genotypes” and the first entry shortened to “Mixed (MM queen, MP workers)”
Changed, thank you.

Figure 1. Suggest revising the y-axis to “Number of workers plus pupae after one year” to indicate measurement unit.
Changed, thank you.

Supplementary material:

Can the authors provide metadata to accompany the supplied dataset and analysis? This would aid with interpreting variables, especially given the occurrence of duplicate variable names.

Given the complexity of the analyses, additional commentary to connect data analysis scripts to specific results would be helpful.

Thank you for this suggestion. We include R markdown scripts and outputs and cleaned data as ESM, which will also be uploaded into dryad.

Reviewers' comments:

Reviewer #1 (Remarks to the Author):

The authors have performed an excellent job in addressing all comments raised by the three reviewers systematically. This has improved the manuscript's focus and clarity significantly. One minor aspect that I am still unsure of is the repeated use of the same colonies for different experiments (or lack thereof). In the rebuttal letter, at page 2, the authors say "For colony size, we included each colony only once in the analyses, when they were one year old." However, in the revised version of the manuscript, the authors say in the methods: "Out of 227 single-queen laboratory colonies, 167 were used only once, and 60 were used more than once. In the statistical analyses, we controlled for having used some laboratory colonies more than once, and for having queens emerging from the same field colonies..." I am therefore confused on what strategies was adopted and what has been accounted for in the statistical analyses.

We thank the reviewer for their comment.

We evaluated colony size when colonies were one year old, thus each laboratory colony was used only once in this analysis. However, we used colonies more than once in the behavioural experiments, and in those analyses, we did control for the colony id. To make this point clear, we have added "in the behavioural experiments" after "we controlled for having used some laboratory colonies more than once" (lines 151-152).

Reviewer #2 (Remarks to the Author):

The authors have adequately addressed all the concerns I previously raised. I recommend the manuscript for publication.

We thank the reviewer for their comment.

Reviewer #3 (Remarks to the Author):

I commend the authors for thoughtfully and thoroughly responding to points raised in the prior review, especially with regards to transparency in data analysis and data sharing. Overall, the relevant background on *F. selysi* is also now much more clearly explained. There are still some minor points of clarification, mostly in the Introduction and Results, that would further improve understanding of the work that was carried out, and of the major findings.

We thank the reviewer for their comment.

Lines 61-64: Couldn't it also be the case that queens of polygyne origin

occasionally mate with males of monogyne origin? But that in field colonies, it is difficult/impossible to distinguish where there is or isn't a mismatch, so as to determine the consequences of either a match or mismatch? (to get at your main point in the paragraph at lines 65-70). I think you do address this more clearly in the paragraph at lines 71-83, so you may want to revise this paragraph accordingly if I have misinterpreted the intention here.

We have better explained the mating system of queens of polygyne origin. They indeed occasionally mate with males of monogyne origin (lines 57-58).

Line 67: To clarify this important point, I'd suggest saying "because queens and workers with each of the alternative supergene genotypes typically develop in separate colonies that differ in social organization and many associated phenotypic traits..."

We have changed this accordingly, thank you (lines 68-69).

Lines 77-80: Modify to "All mature colonies typically contain..." so this does not contradict other sections of the Introduction.

We have changed this accordingly, thank you (line 78).

Line 159: suggest revising to , "...and eggs and larvae were not counted."

We have changed this accordingly, thank you (line 160).

Line 175: suggest revising to "We compared the size of colonies (workers plus pupae) with monogyne, polygyne, and "mixed"..."

We have changed this accordingly, thank you (line 176).

Line 303: revise to "Overall, queens in colonies containing workers with the paternally-inherited P haplotype were more likely to die...". Then at line 304, "Specifically, queens..."

We have changed this accordingly, thank you (lines 304-305).

Line 309: For clarity, revise to "...less likely to die than polygyne queens in colonies with only polygyne workers"

We have included this change with a minor alteration for consistency, thank you (line 309-310).

Figure 3: Is it appropriate to have the points joined together with gray lines?

We have changed the figure accordingly.

Line 435: Might be useful to point out that this spread may potentially be offset by the tendency of monogyne queens to mate assortatively, and/or have colonies that grow more quickly. (to explain the overall persistence of both social forms across the species range)

We have included this point with slight modifications, thank you (lines 434-438).